# Construction of the Uyghur Noun Morphological Re-Inflection Model Based on Hybrid Strategy

**Muhetaer Munire [1,2,3], Xiao Li [1,2,3,*] and Yating Yang [1,2,3]**

1 Xinjiang Technical Institute of Physics & Chemistry, Chinese Academy of Sciences, Urumqi 830011, China; munire@ms.xjb.ac.cn (M.M.); yangyt@ms.xjb.ac.cn (Y.Y.)
2 University of Chinese Academy of Sciences, Beijing 100049, China
3 Xinjiang Laboratory of Minority Speech and Language Information Processing, Urumqi 830011, China
* Correspondence: xiaoli@ms.xjb.ac.cn; Tel.: +86-136-0993-8871

**Abstract:** In this paper, a hybrid strategy of rules and statistics is employed to implement the Uyghur Noun Re-inflection model. More specifically, completed Uyghur sentences are taken as an input, and these Uyghur sentences are marked with part of speech tagging, and the nouns in the sentences remain the form of the stem. In this model, relevant linguistic rules and statistical algorithms are used to find the most probable noun suffixes and output the Uyghur sentences after the nouns are re-inflected. With rules of linguistics artificially summed up, the training corpora are formed by the human–machine exchange. The final experimental result shows that the Uyghur morphological re-inflection model is of high performance and can be applied to various fields of natural language processing, such as Uyghur machine translation and natural language generation.

**Keywords:** uyghur noun; hybrid strategy; morphological re-inflection; machine learning

## 1. Introduction

Highly morphological languages still pose a challenge for natural language processing (NLP). Due to highly inflected word forms in the Uyghur language, the data are sparse causing otherwise state-of-the-art systems to perform poorly on standard tasks, such as machine translation and question answering, where errors in the understanding of morphological forms can siginifcantly harm the performance [1]. Accordingly, learning morphological inflection patterns from labeled data is an important challenge. To create systems whose performance is not deterred by complex morphology, the development of NLP tools for the generation and analysis of morphological forms is crucial. In addition, applying inflection generation as a post-processing step has been shown beneficial to alleviating the data sparsity when translating highly morphological language [2]. Many languages including Uyghur have complex morphology with dozens of different word-forms for given target tags and lemmas, for example, kitabiglar. In this case, the second person plural presenting the tense of kitabiglar is generated. To design an Uyghur Noun Re-inflection model (UMR), it is possible to follow a linguistic approach by using prior knowledge of the morphology of the Uyghur language, or a statistical approach based on statistical principles that infer the word formation rules from the corpus of documents in the language [3].

The words in the Uyghur language are divided into 12 categories. The number of suffixes is more, and the concatenation rules between the stem and suffix of each categories are different and complicated [4]. According to the statistics of the frequency of various categories of words in the Uyghur million-word corpus, the most commonly used category is noun. Therefore, by implementing the Uyghur Morphological Re-inflection model of noun class, this article introduces the pre-work before the realization of the model [5].

## 2. The Nominal Paradigm of Uyghur Language

Uyghur text is written as it is pronounced, with each phoneme recorded by a character. Therefore, a total of 32 characters correspond to 32 phonemes (8 vowels and 24 consonants) [6]. Uyghur nouns are a modifiable word class. In Uyghur, there are 49 noun suffixes, two times more than those in Turkish. These suffixes can be divided into three categories, namely, Number category, Ownership-dependent category and Case category [7].

### 2.1. The Singular and Plural Forms of Uyghur Noun

When Uyghur nouns go into a sentence, they either occur in the singular form, which indicates a singular concept, or in the plural form, which indicates a plural concept. According to the Uyghur Noun's word-formation rule, the plural suffixes only appear after the noun stems, and the rest of the suffixes of Noun class cannot appear in front of the plural suffixes [8]. For example, the plural suffixes "لار/لەر" are chosen based on the phonetic harmony rule between stem and suffixes.

### 2.2. The Ownership-Dependent Forms of the Uyghur Noun

The ownership-dependent category of the noun is a grammatical category. In Uyghur language, this category is expressed by the possessive forms that are made by adding the noun's ownership-dependent suffixes [9]. The ownership category can be connected directly behind the stem or the plural suffixes. The various ownership-dependent forms of the noun and details of their formation may be seen in the following Table 1 [10].

**Table 1.** Suffixes of the ownership category.

| Type | | Suffix | Examples | In English |
|---|---|---|---|---|
| 1st person | Singular type | م،ىم،ۇم،ۈم | بالام،قەلىمىم،قولۇم،كۆزۈم | My child, My pen |
| | Plural type | مىز،ىمىز | بالىمىز ،قەلىمىمىز | Our children |
| 2nd person | Singular ordinary type | ڭ،ىڭ،ۇڭ،ۈڭ | بالاڭ،قەلىمىڭ،قولۇڭ | Your child, |
| | Plural ordinary | ۇڭلار،ڭلار ،ىڭلار ،ۈڭلار | بالاڭلار، قەلىمىڭلار | Your children |
| | Singular refined type | ڭىز،ىڭىز | بالىڭىز، قەلىمىڭىز | Your child |
| | Singular and plural respectful | لىرى | بالىلىرى | His/their children |
| 3rd person | | سى،ى | بالىسى،قەلىمى | His/their pen |

### 2.3. The Case Category of the Uyghur Noun

The case category of the noun is expressed in the Uyghur language by means of case forms which are formed by adding nominal case suffixes. These suffixes can be directly attached to the noun stem [11]. If there is another noun suffix, case suffixes must be attached to the outermost. The case of Uyghur nouns is divided into ten varieties as the exposition above. The details centering on the formation of these case forms are shown in Table 2.

**Table 2.** Suffixes of the case category.

| Case Name | Case Suffixes | Examples | In English |
|---|---|---|---|
| Nominative case | Null | مەيدان،كىتاب،ئۆي،دەرس | Square, book, house, lesson |
| Possessive case | ـنىڭ | كىتابنىڭ | Book's |
| Dative case | ـغا،ـقا،ـگە،ـكە | سىرىتقا، ئۆيگە،دەرسكە  ,مەيدانغا | To square, To outside, |
| Accusative case | ـنى | كىتابنى | This book |
| Locative case | ـدا،ـتا،ـدە،ـتە | مەيداندا،كىتابتا،ئۆيىدە،دەرستە | On the square/…book |
| Ablative case | ـدىن،ـتىن | ـمەيداندىن،ـكىتابتىن | From square, from sth |
| Locative-qualitative | ـدىكى،ـتىكى | ،باغدىكى،شەھەردىكى | Flower (grows in gardens) |
| Limitative case | غىچە،قىچە،كىچە،گىچە | سىنتەبىرگىچە،قۇ لاقلىرىغىچە | Till the September |
| Similitude case | ـدەك،ـتەك | ـبالاڭدەك،كىتابتەك | Like your child |
| Equivalence case | ـچىلىك،ـچە | ـمەيدانچىلىك،ـكىتابچە | As a square |

### 3. The Interconnection Rules of Uyghur Noun Stem and Suffixes

In Uyghur, there are multi-variant suffixes with different variants of one suffix added to harmonize the phonetic characteristics of the particular stem to which suffix is added [12]. The Nominal Paradigm of Uyghur Noun is:

$$\text{STEM + [Number] + [Ownership] + [Case]}$$

A stem (or root) is followed by zero to many suffixes, to stem a word is to reduce it to the root of the word. There are two number suffixes, 23 ownership dependency suffixes, and 24 case suffixes for Uyghur nouns, four times those in Turkish [13]. According to the concatenation rule of the three category suffixes, for a single entry in the lexicon, which is for a single noun stem, there are plenty of possible inflections: $C_3^1$ (the 2 Number suffixes + the Number free form) for number times, $C_{24}^1$ (the 23 Ownership suffixes + the Ownership free form) for possession times and $C_{25}^1$ (the 24 case suffixes + the case free form) for case inflection. The expression is shown in the following Equation (1):

$$\text{Noun} = C_{n_1}^1 C_{n_2}^1 C_{n_3}^1 (n_1 = 3,\ n_2 = 24, n_3 = 25) \tag{1}$$

Hypothetically, there are four noun stems in a sentence, and the possibilities of connecting each noun stem and suffixes are 1800. Naturally, things get further complicated. We will eventually get a particularly large number of sentences, as shown in the Equation (2):

$$\prod_{i=1}^{N_i} C_{count_i}^1\ count_i = 1800 \tag{2}$$

Ni represents a number of nouns in the sentences. If *i* = 4, based on the formula, $1.0497 \times 10^1$ possibilities of sentences are generated. Obviously, this is a huge number, so we have to use some methods to reduce the total number of sentences that are ultimately generated to compose the best sentences. In the next few sections, we will discuss certain methods one by one.

### 4. Description of Morphological Re-Inflection Model

The entire structure of Uyghur Nouns Re-inflection (UNR) model is composed of several approaches and realized by the perfect combination of several modules, such as data preprocessing [14], training, and testing modules, each of which has an irreplaceable role. In the next few sections, we will introduce those modules, respectively.

### 4.1. Design for Data Structure

The data structure consists of a nested hash table. In computing, a hash table is a data structure which implements an associative array of abstract data type, a structure that can map keys to values [15]. A hash table uses a hash function to compute an index into an array of buckets or slots, from which the desired value can be found. However, the nested hash table to be described in this article is a structure in which the value of a hash table is equal to another hash table [16]. Figure 1 depicts a configuration diagram of a nested hash table.

According to Figure 1, the key on the left side is the key to the top hash table. Its value is equal to another hash table on the right side which we call an internal hash table. The key is the internal hash table's key, and its value is the frequency of every Key in the entire data. To obtain a clearer expression, it is necessary to explain that the top-level hash table is composed of the n-gram sequences of the noun stem (denoted by N) and neighbor words (denoted by W1 and W2), while the hash value of the top-level hash table contains another internal hash table. It is the hash key for the n-gram vectors of nouns plus suffixes (denoted by N+ Suffix) and neighboring words. The hash value of the internal hash table is the statistical probability of the internal hash table key. The following is the data format description of this nested hash table:

$$\{W_1 \mid N \mid W_2 : \{\ W_1\ \mid N + \text{Suffix} \mid W_2 : \text{PRO}\}\}$$

In this expression, W indicates normal words with suffix, N stands for noun stem, N+ Suffix denotes noun stem plus its suffix, PRO represents the probability of the combination of N+ Suffix and its neighbor W.

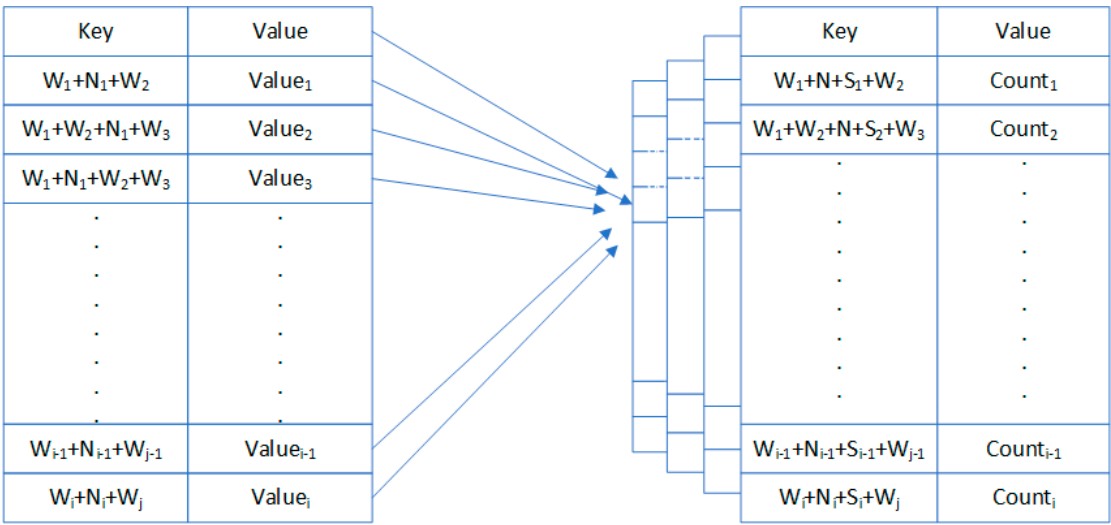

**Figure 1.** Diagram of a nested hash table.

## 4.2. Data Preprocessing

Data preprocessing refers to some processing of data before the main processing. This is necessary because the object of re-inflection that we discuss in this article should appear as the noun stem form with part-of-speech, and all words in the sentences except noun must remain in their original form without part-of-speech to get the exact format of the data. First, identify the part of speech of each word in every sentence. At present, the Uyghur part-of-speech tagging tools which were developed previously can label not only the noun stem [17] but also each component of the sentences after stemming suffix segmentation (shown in Example 1). However, to the data structures, those we will actually use in this research, partial stems and suffix segmentation and POS tagging (part-of-speech tagging, is the process of marking up the words in a text) are unnecessary and seriously influence the quality of the UNR. Therefore, considering the highly agglutinative features of Uyghur language and disadvantages of the fewer data resources, we have changed the original data corpus as shown in Example 2, format of the training data shown in Example 3, and test data shown in Example 4. The following shows the normal form of one example that is randomly extracted from the data corpus. The data with POS tagging comes from the original data as the following Example 1. After some processing, the data of this sentence and its changed forms have been obtained.

Example 1: Non-POS tagging sample of the sentence

In Uyghur language: هۆكۈمەت خىزمىتىنى كۈچەيتىش

In English: Strengthening government work

Example 2: POS tagging sample of the sentence

هۆكۈمەت|هۆكۈمەت N | نى+ى+خىزمەت|خىزمىتىنى N+PERS+CASE ش+كۈچەيت|كۈچەيتىش | Y |؛ V+VERB_GER

In Example 2, the words marked with green are the original words. The noun stem-suffix and the labeling which are marked red are the data which we use in this research while the rest of the other colors are non-noun classes and stems and their POS tagging, which are unnecessary.

We use the function of conversion to modify the corpus to get the training data corpus and test data corpus with different data forms. The form of the training data that we have is shown in Example 3:

كۈچەيتىش نى+ى+NN ؛

The form of the test data that we have is shown in Example 4:

<div dir="rtl">كوٚچهيتنشNخىزمەت| Nھوٚكوٚمەت|</div>

## 4.3. Data Training

According to the given training data, first, obtain the noun stem and its neighbor word in each sentence. Based on the N-gram combination method of taking the noun stem as the center, choose the first three adjacent words from the right side and left side of the all noun stems N+ suffix, respectively. Second, constitute a combined set which contains at least two words (one N+ suffix and one word), but not more than four words (one N+ suffix and three words). Clearly, the window size of N-gram sequence combination value ranges from two to four. Third, calculate the probability of each combination in the entire training corpus before sending them to an internal hash table which we have described above. For example, there are the data as follows:

$$W_i \mid N+Suffix \quad W_1 \mid N+Suffix \mid W_2 \quad W_j$$

And the part of the composition of the N-gram sequence is as follows: following a1, a(n-1), and an are N-centered combinations that take the neighbor words from the N's left side; b1, b(n-1) and bn are also N-centered combination but take the neighbor words from the N's right side, as are indicated through the following combination example:

$$a_1 = W_1 \mid N + Suffix \; a_{(n-1)} = W_{i+1} \mid W_1 \mid N + Suffix \; b_{(n-1)} = N + Suffix \mid W_2 \mid W_{2+j}$$

$$b_n = N + Suffix \mid W_{j-2} \mid W_{j-1} \mid W_j \; a = W_i \mid W_{i-2} \mid W_1 \mid N + Suffix \; (2 \leq N \leq 4)$$

Wi, Wj and all other W with different indexes are the adjacent words. N stands for the noun stem, Suffix refers to the noun suffix. In each parenthesis is an N-gram sequence combination. It is worth mentioning that there are many words in the Uyghur language. Dozens of non-repetitive nouns may appear in text data, which leads to the value of the internal hash table being equal to zero. To avoid zero probability, which influences the test results, all the noun stems are replaced by N, and the noun suffixes are retained and separated out as "+" from the *N*. All other words containing actual word forms in the text are not replaced by *W*. After obtaining the probabilities of each combination, *N* is replaced by its actual noun stem. Subsequently, in the test module, the matched noun stems and word suffix are verified. We represent all of the combinations with *C* and use the following Equations (3) and (4) to calculate the probability of each combination:

$$CTotal = \sum_i^j a_{i,j} \tag{3}$$

$$P_{a_{i,j}} = C_{Total} / a_{i,j} \tag{4}$$

CTotal is the total number of the whole combination while $a_{i,j}$, $P_{a_{i,j}}$ is the probability of each combination in the file.

## 4.4. Testing

The testing module is one of the crucial modules of the UNR, with the main processing of this module including the two steps as follows:

### 4.4.1. Get the Test Data Combination

The test data is based on the given test data to obtain the *N*-gram sequence combination that includes noun stem *N* and its neighbor words (the window size is two to four). As mentioned in the previous section, our data structure is made up of nested hash tables, and the test data is in the top (external) hash table, with the hash key being the sequence of a combination containing noun

stems and its neighbor words as we have mentioned before. The hash value is training data which are equal to another hash table (internal). Our re-inflection model including the top hash table key and value obtains all of possible <N+ suffix> structure sequences that match the present test data sequence combination, then chooses the suffixes with the highest probability.

### 4.4.2. Suffix Selection

In Uyghur sentences, suffixes of nouns are determined by the words at the end of the sentence. However, most Uyghur sentences end with verbs. But the Uyghur sentence components have their own diversity, giving rise to some special cases. Therefore, we cannot completely neglect the possibility of some words being closer to the current words which can also determine the possibility of noun class suffixes. For example, if there are three N-gram sequence combinations and each one has their probability as in the following Table 3:

**Table 3.** Part of the information about combination details.

| | | Gram | 2 | 3 | 4 |
|---|---|---|---|---|---|
| شیاۇمیي\<XiaoMei> | | probability | 0.0093 | 2.648 | 4.56 |
| | | combination | ['N', 'N'] | ['N', 'بىر', 'ى'] <N,N+one,suffix> | ['N+دا', 'پىيىم', 'بىر', 'دىن'] <N+suffix,N+one,suffix+handcapes> |
| | | position | 0 | 0 | 0 |
| نىسىملىك\<name> | | Gram | 2 | 3 | 4 |
| | | probablity | 0.00015 | 4.5617 | 4.56173 |
| | | combination | ['N', 'بىر'] <N,one> | ['N+پىيىم', 'بىر', 'دىن'] <N+handcapes,one,suffix> | ['N+پىيىم', 'بىر', 'دىن', 'N'] <N+handcapes,one,suffix,N> |
| | | position | 0 | 0 | 0 |
| قىزچاق\<girl> | | Gram | 2 | 3 | 4 |
| | | probablity | 1.80188 | 6.64873 | 4.56173 |
| | | combination | ['N', 'يازغان'] <N,wrote> | ['N+يازغان', 'غا', 'لىرى', 'N'] <N+wrote,suffix+suffix,N> | ['N+يازغان', 'نى', 'لەر', 'ئ', 'N'] <N+wrote,suffix+suffix,N> |
| | | position | 0 | 0 | 1 |

Considering the special situation of Uyghur language represented above, the 4-gram combination is supposed to be the farthest combination of the whole N-gram sequence combination. That is why we start with 4-gram combination to search for the best suffixes, find the highest probability of the 4-gram combination, and take them as the best suffixes. If there is not a 4-gram combination of the data, start with 3-gram sequence combination before taking the one which has the highest probability of their combination. If there is no 3-gram combination, the same searching process will start as usual. This gradual search will continue until the best combination with the highest probability is found. Then optimal combinations will be used as the value of the external hash table. Based on that value, find the key to the external hash table N-gram data sequences. After these processes, the best suffixes of the noun stem are finally obtained. Then, replace the N with the actual noun stem. In the end, UNR contains the selected stem concatenate with the noun stem. As for how to connect the stem and suffix together, the details are introduced in the next section.

## 5. Matching of Uyghur Stem and Suffix

The fifth section is composed of four subsections. The first subsection introduces all variants of Uyghur noun affixes. The second subsection links the classification of Uyghur alphabets. The third subsection details the previous two subsections for reference. The stem affix link rules, the last section

introduces the Uyghur phonetic changes that occur after the stem affixes are connected according to the link rules in the previous section. The content of each section in this section is related to the previous section.

### 5.1. Uyghur Noun Suffix Variants

As mentioned in the previous section, all noun stems in the training set are replaced by N first, and then, after the UNR finds the best suffix through probability calculation, all N will be replaced by the original noun stems (actual word stem), thereafter is the step of the connection rules for the actual noun stem and its corresponding suffix which UNR matched previously. As we all know, in Uyghur language, stem and suffix are made up of one or several letters, the connection between stem and suffix can be seen as the interconnection of two letters, but this is not a simple letter connection. Each connection should follow the rules of Uyghur phonetic harmonization, and all the phonetic changes that occur during phonetic harmony should be satisfied.

Most of the suffixes in the Uyghur language have their different variants, and noun suffix is divided into 16 categories. Among them, there are two variants for each of these seven categories, six variants for each of the six categories, and none for the rest. Obviously, different variants are composed of different vowels and consonants. All the vowels and consonants in the Uyghur language have their certain matching rules, and the structure is very complicated. This article refers to the Uyghur morphological related knowledge summed up the series rules that different stem which ends of a different letter can be connected with which suffix or not. The number of all suffixes in the Uyghur language is limited, but the stem is unlimited, and each of the 32 Uyghur letters can be the ending letter of a noun stem. The purpose of the rule base is to identify the letters contained in the noun stem ending with different letters to select which one of the suffix variants matches the correct stem. As long as we find the matching rules of each Uyghur letter and noun suffix, we can then verify whether the suffix affirmed by UNR is correct or not. If it is not, then gradually search and match the corresponding suffix.

### 5.2. Uyghur Alphabet Classification

Uyghur language has eight vowels. According to the pronunciation of the tongue surface of different parts of pronunciation, the eight vowels can be divided into three types such as a front vowel, middle vowel, and back vowel. According to different lip shapes when pronounced, the eight vowels can also be divided into rounded vowels and exhibition vowels [18]. Table 4 shows the classification of Uyghur vowels based on the pronunciation method.

**Table 4.** Uyghur vowel classification.

| Vowel Type | Front Vowel | Middle Vowel | Back Vowel |
|---|---|---|---|
| Rounded vowel | ئۆ ئۆّ | | ئۇ ئو |
| Exhibition vowel | ئە | ئې ئى | ئا |

Uyghur language has 24 consonants. According to the vocal sound, the 24 consonant letters can be divided into voiceless consonants and voice consonants. Table 5 shows the classification of Uyghur 24 consonants [19].

**Table 5.** Consonants classification.

| voice consonants | ؤ | گ | ژ | ي | ن | م | ل | غ | ز | ر | د | ج | ب | ھ |
|---|---|---|---|---|---|---|---|---|---|---|---|---|---|---|
| voiceless consonants | ق | چ | پ | ك | ف | ش | س | خ | ت | | | | | |

### 5.3. Construction of Variant Matching Rules Repository

One of the most important parts of this research is how to find the correct suffix from several noun suffixes which match an unlimited number of Uyghur noun stems. After a series of studies on Uyghur alphabet collocation, this paper has summarized the following five rules:

#### 5.3.1. Voice Consonants Stemmed Noun Suffixes

1. Except for all stems ending with consonants <ب> and <د> that match only the complex suffix <لار>, most of the other stems ending with voiced consonants match the complex suffix phase <لار>, <لەر>.

2. All stems ending with voice consonants match the ownership of suffix variants <ى><ىڭىز><ڭىز لا><لىڭ><مسز><مم> which begin with the vowel.

3. Most of stems ending with voices consonants match the ownership of suffix variants <كچە><غچە><دىكى><دەك><دىن><دە><دا><ىگە><غا>, where the stem ending with consonant <د> match all the limitative cases of suffix and suffix variants <تىكى><تەك><تىن><تە><تا>, and the stem ending with consonant <ج> match the suffix variants <كچە><كە>.

#### 5.3.2. Voiceless Consonant Stems and Noun Suffixes

The stem ending with a voiceless consonant matches all the suffix variants that begin with a vowel and all the suffix variants that begin with a voiceless consonant. However, the stem ending with consonant <ڭە> only matches the suffix variants <مسز><ىڭلار><لىڭ><مم> which start with partial vowels but still matches all the suffix variants that begin with a voiceless consonant.

#### 5.3.3. Front Vowel and Noun Suffixes

Mostly, stems ending with a front vowel match the suffix <دىن><دەك><دىكى><گىچە><لەر><م><مسز><ىڭلار><ىڭز><ىگە><دە> with variants starting with the voice consonants. However, some of the special cases are that the suffix variants <سى> which start with a voiceless consonant that can still match the stem which ends with the front vowel.

#### 5.3.4. Back Vowel and Noun Suffixes

Mostly, stems ending with back vowel match the suffix variants <ىڭز><غا><دا><دىن> <غىچە><لار><م><مسز><ىڭلار> that start with voice consonants. However, some of the special cases are that of suffix variants <لار> starting with a voiceless consonant that can still match the stem which ends with the back vowel.

#### 5.3.5. Middle Vowel and Noun Suffixes

In most cases, stems ending with middle vowel match the suffix <دىن><گىچە> <غىچە><لار><م><مسز><ىڭز><لار>< ىڭز><دەك><دىكى><غا>< گە><دا >< دە> variants. However, in some special cases, suffix variants start with a voiceless consonant. The following is a practical example to illustrate the selectivity of the noun suffix variants that have been described above. Example A contains three re-inflection objects whose stems are re-inflected are shown in Table 6. And the blue cells in the table are the final <matching group> found by UNR that includes the noun suffix plus N which replaces the noun stem. The green part is the actual stem forms without N plus suffix while the yellow part is the final choice of the noun suffix. However, the cells marked with red color must be the correct suffix variants that are consistent with all these rules discussed above. The last grey part shows the ultimately correct version of the suffix corresponding to the current stems.

A ‫ئۇ N+دىن N+غا،قاراپ ماڭدى،ھەمدە N+غىچە كىبەلمەيدىغانلىقىنى ئېيتىپ بىز بىلەن خوشلاشتى.‬

**Table 6.** Part of example of the noun stem variants.

| N+Suffix | Replace the Actual Stem | | Suffix Variants | | Stem Correct Suffix |
|---|---|---|---|---|---|
| N+دىن | مەكتەپ+دىن | دىن | تىن | | مەكتەپ+تىن |
| N+غا | سەرت+غا | گە غا | قا | كە | سەرت+قا |
| N+غىچە | سىنتەبىر+غىچە | قىچە غىچە | گىچە | كىچە | سىنتەبىر+گىچە |

## 5.4. Uyghur Phonological Phenomenon

When the stem and suffix are concatenate with each other, the surface forms often change (or harmonize) the boundary according to certain phonetic rules. Phonetic change (harmony) is the basic controlling rule in the root-suffix linkage. There are two types of phonetic change (harmony) in Uyghur language: consonant harmony and vowel harmony. When certain morphemes are concatenated, the last vowel of the previous morpheme is harmonized with the first vowel of the next morpheme according to their tongue position [20]. This phenomena is called vowel harmony. Similarly, the last consonant of the previous morpheme is harmonized with the first consonant of the next morpheme according to their voice or voiceless character. This phenomena is called final consonant harmony. Phonetic harmony is a complex phenomenon. There are four types of changes (harmony) which cause different surface forms of morphemes as shown in Table 7.

**Table 7.** Uyghur morphological changing phenomenon.

| Weakening | | Insertion | Deletion | Weakening/Deletion |
|---|---|---|---|---|
| **Vowel's Weakening** | **Consonant's Weakening** | | | |
| قەلەم+ىم=قەلمىم | كەل+ىپ=ىدىم=كېلىۋېدىم | ئارزۇ+ئۇم=ئارزۇيۇم | بۇرۇن+ى=بۇرنى | چال+ىپ+تۇ+كەن+چىلىپتىكەن |
| vowelئاturn to vowelئى | Consonantپturn to ۋ | Adding a consonantيafter the stem | Second consonant تۇ in the stem have deleted | ئى،ئۇ turn toل،ئىturn to and first ئائ has deleted |

As seen from the Table 7, there are some phonetic changes when the noun stem and suffix are connected with each other. Except for some other word classes which do not follow the rules of phonetic changes, most of noun stems and suffixes follow the rules when they are connected with each other. Therefore, when the UNR connects certain noun stems and suffixes, those cases that do not follow the rules can be negligible. It is more reliable to follow all Uyghur phonetic change (harmonize) rules.

## 6. Experiment

### 6.1. Design of the Experiment

We have prepared a corpus of 100,000 and 1,000,000 Uyghur sentences to carry out experiments. The previous works showed that the performance of learners can benefit significantly from much larger training sets [21–29]. The corpus covers the fields of news, entertainment, and sports. Each sentence is complete, and each term has been carried out in different forms of part-of-speech tagging, in which each of training set and test set has different forms of tagging, respectively. In this paper, 5% of two different statements are extracted randomly as the development set and test set, and the remaining 90% of the sentences are used as the training set. At the end, the re-inflection efficiency of UNR in 100,000, 500,000, and 1 million Uyghur sentences are compared, respectively.

### 6.2. Experiment and Analysis

Experiment: Experiments on 100,000, 500,000, and 1 million Uyghur sentences
Purpose: to implement the noun forms morphological Re-inflection

Focus: Regardless of other parts of speech, examine the UNR Re-inflection efficiency to noun stems
Evaluation criteria: similarity, recall, accuracy, F-measure.

The similarity described in this article refers to the String metric, a string metric (also known as a string similarity metric or string distance function) is a metric that measures the distance between two text strings for approximate string matching or comparison and in fuzzy string searching. For example, the strings "Sam" and "Samuel" can be considered to be close. A string metric provides a number indicating an algorithm-specific indication of distance.

Table 8 displays the experimental data result of the similarity of the strings on Train set 24.1 MB, 128 MB, and 249.6 MB, respectively. It can be easily observed that the similarity of the 100,000 data corpus is approximately 95.08. The value of similarity increases gradually as the date corpus size increases. When the data size is around 500,000 sentences, the similarity of the strings is about 95.38. When the data size is around 1 million sentences, the similarity of the strings is about 95.60.

**Table 8.** Similarity of the string character.

| Data Amount | Measure | |
|---|---|---|
| Train set | Test set | similar |
| 24.1 MB | 66 KB | 95.08 |
| 128 MB | 66 KB | 95.38 |
| 249.6 MB | 66 KB | 95.60 |

The results of the correlational analysis are set out in Table 8 and Figure 2. It can be seen that there is a significant difference between the results of the groups of data, which are 0.3 and 0.22, respectively.

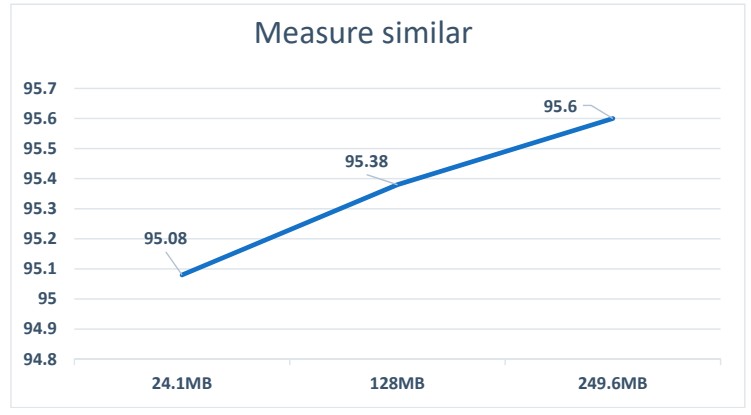

**Figure 2.** Comparison of similarity among different training datasets.

It can be seen from the data in Table 9 that the values of the accuracy, recall, and F-measure of the about 100,000 sentences with the size of 24.1 MB are approximately 63.10, 66.56, and 64.78, respectively. The values of the accuracy, recall, and F-measure of about 500,000 sentences with the size of 128 MB are approximately 64.65, 69.91, and 67.18, respectively. The values of the accuracy, recall, and F-measure of the about 1million sentences with the size of 249.6 MB are approximately 66.59, 70.06, and 68.28, respectively.

**Table 9.** Experimental results of all measure.

| Data Amount | | All Measure | | |
|---|---|---|---|---|
| Train set | Test set | accuracy | recall | F-measure |
| 24.1 MB | 66 KB | 63.10 | 66.56 | 64.78 |
| 128 MB | 66 KB | 64.65 | 69.91 | 67.18 |
| 249.6 MB | 66 KB | 66.59 | 70.06 | 68.28 |

Figures 2 and 3 provides the correlations among all measures of the UNR experiments. It can be seen that there is a significant difference between the results of different groups of measures, which are 1.55, and 1.94, respectively on the measure of accuracy, 3.35 and 0.15, respectively on the measure of recall, and 2.4 and 1.1, respectively, on the F-measure. All measures gradually increase with the increase of the corpus size.

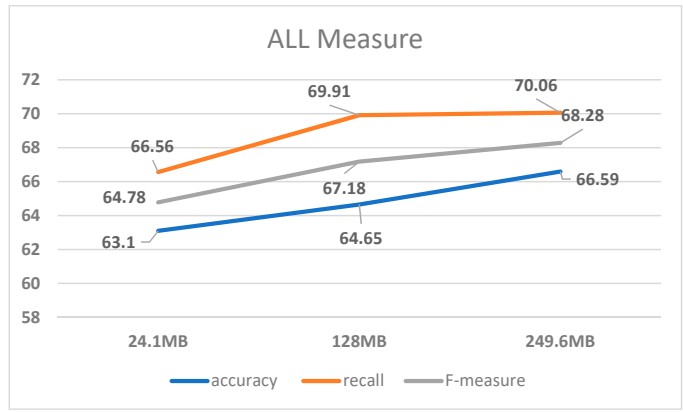

**Figure 3.** Comparison of accuracy, recall and F-measure among different training datasets.

## 7. Conclusions

In this paper, through a series of experiments, a combination of rules and statistical methods and the realization of a single sentence in the noun stem reduction model can be seen. The Uyghur morphology reduction model performance is quite good, of course, as used in this article. The limitations and errors of the Uyghur stem-word affixing annotation tools, as well as collocation rules of some foreign words, lie in that they are not in line with the lexical and grammatical rules we have collected. For the purposes of this paper, we have increased the size of the corpus as much as possible, as the Uyghur language has a small corpus. Given that it has extremely complicated lexical and grammatical rules, we are pleased that it can be observed from the experiments conducted that the recognition efficiency of this model is quite satisfactory. We can predict that, in the course of further research, the UNR performance quality will be improved through targeted optimization of inaccuracies in labeling, which will affect overall UNR performance and incompleteness of grammar rules.

**Author Contributions:** M.M., X.L. and Y.Y. conceived the algorithm, prepared the datasets, and wrote the manuscript. All authors read and approved the final manuscript.

**Funding:** This work is supported in part by the West Light Foundation of The Chinese Academy of Sciences (Grant No. 2017-XBQNXZ-A-005), The National Natural Science Foundation of China (Grant No. U1703133), Youth Innovation Promotion Association CAS (Grant No. 2017472), The Xinjiang Science and Technology Major Project under (Grant No. 2016A03007-3), The High-level talent introduction project in Xinjiang Uyghur Autonomous Region (Grant No. Y839031201).

**Acknowledgments:** The authors would like to thank all anonymous reviewers for their constructive advices.

**Conflicts of Interest:** The authors declare no conflict of interest.

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
