# Peer review of "Construction of the Uyghur Noun Morphological Re-Inflection Model Based on Hybrid Strategy"

_applsci, doi:10.3390/app9040722_

Round 1

Reviewer 1 Report

The article presents a system for morphological reinflection of nouns in the Uyghur language. Morphological reinflection is useful for systems that have to provide output in the language, e.g. natural language generation or machine translation. The Uyghur language is especially challenging in this respect, due to being morphologically rich and agglutinative, having a variety of affixes and sandhi phenomena.

The presented model is a traditional approach based on prior linguistic knowledge, and probabilities estimated from counts on a corpus. While it is not an especially novel approach, the application to Uyghur is interesting as not much work of this kind has been done on this language. However, it would be interesting to contrast this approach to models based on machine learning.

The paper is reasonably clear, except for some issues pointed out below. However, the paper should go through proof-reading before final publication as there are some grammatical errors and odd English constructions (some examples in the abstract: "are remained in the form of stem" should be "are given as stems", "the noun are re-inflected" to "the nouns are re-inflected", etc.)

Some minor comments:

- In lines 88-91, please explain what is the relevance of the number of possible sentences. In principle each noun could be treated in isolation, so it is not clear why this is relevant.

- In Table 3, please add English translations. While other examples used in the paper have translations, the examples in this table are difficult to understand for non-Uyghur speakers.

- In Section 6.2, it is not clear what the "similarity" metric is. Please add a definition of this metric.

- Also in Section 6.2, the experiments should compare to a baseline. A naive baseline that can be used is inflecting each noun to the most common form seen for that noun in the corpus.

Author Response

The point-by-point response to the reviewer’s comments can be found in attached word file. Thanks!

Reviewer 2 Report

In this manuscript, a hybrid strategy of rules and statistics is proposed to implement the Uyghur Noun Re-inflection model. More specifically, relevant linguistic rules and statistical algorithms are used to find the most probable noun suffixes and output the Uyghur sentences after the noun are re-inflected. The experimental results on several datasets demonstrated that the proposed method is of high performance and can be applied to various fields of natural language processing. Generally speaking, the proposed manuscript is well organized and easy to read. I have the following comments:

       Although the level of language in the manuscript is sufficient, there are still minor flaws. In order to make it easier for readers to better understand the objectives and results of the study, the authors need to carefully polish it.

2.       Although the general idea of section 5 is understandable, the three subsection 5.1, 5.2, and 5.3 are somewhat isolated from each other. Instead of just generally introducing them separately, the author needs to use a flowchart or a paragraph to describe how they coherently work together in the revised manuscript.

3.       The interpretation of the formula 1 is not very clear. Please clearly explain the meaning of symbols.

4.       Please rewrite the abstract section.

5.       The author omitted some recent references in the manuscript.

6.       The figures are too blurred to read(Figure 1, 2). High-quality figures should be provided.  

Author Response

(The authors gave the same response as above.)
